# Diabetes-Related Knowledge, Attitude, and Practice Towards Exercise and Its Benefits Among Individuals with Type 2 Diabetes Mellitus

**DOI:** 10.3390/ijerph21111529

**Published:** 2024-11-18

**Authors:** Nokuthula Mtshali, Takshita Sookan-Kassie

**Affiliations:** Discipline of Biokinetics, Exercise and Leisure Sciences, School of Health Sciences, University of KwaZulu-Natal, Durban 4000, South Africa; sookan@ukzn.ac.za

**Keywords:** knowledge, attitude, practice, physical activity, type 2 diabetes mellitus

## Abstract

Regular exercise has been shown to have a positive effect on the health outcomes of individuals with type 2 diabetes mellitus (T2DM); however, it is still underutilized in developing countries. This study investigated diabetes-related knowledge, attitudes, and practice toward exercise and its benefits among individuals with T2DM. A total of one hundred and ninety-nine participants (male = 21.6% and female = 78.4%) with T2DM, aged between 18 and 75, (43.77 SD± 14.78) years, participated in the study. A validated diabetes-related knowledge, attitude, and practice questionnaire, evaluated by true/false or unsure, was utilized in the study. Descriptive and inferential statistics analysis was used to analyze data with the significance set at *p* < 0.05. The results demonstrated poor diabetes-related knowledge of exercise (*p* < 0.001). The majority of the study participants, 163 (81.90%) with T2DM, did not know that physical activity and exercise are different. About 158 (70.40%) of the study participants felt that their regular work was an adequate substitution for exercise. More than 50% of the study participants demonstrated a positive practice towards exercise and its benefits towards T2DM (*p* < 0.001). The majority of the study participants reported poor knowledge and a negative attitude toward diabetes. However, their diabetes-related practices were satisfactory.

## 1. Introduction

Diabetes mellitus is a metabolic disorder characterized by insulin deficiency, which results in chronic hyperglycemia [1]. It is one of the top 10 causes of death in adults, estimated to have caused 4 million deaths globally in 2017, and this number is estimated to reach 700 million by 2045 [1]. There are three main classifications of diabetes mellitus, namely type 1 diabetes (T1DM), type 2 diabetes mellitus (T2DM), and gestational diabetes mellitus (GDM). Since 2000, the International Diabetes Federation (IDF) has reported on the national, regional, and global occurrences of diabetes, and T2DM [2,3], which accounts for about 90% of cases [4], is experiencing a rising trend due to factors such as the persistent hyperglycemic environment, which significantly enhances the production of advanced glycation end products (AGEs) through various biochemical pathways [5]. The resultant accumulation of AGEs leads to oxidative stress and cellular dysfunction, contributing to the pathogenesis of multiple chronic diseases associated with aging, obesity, and metabolic disorders [5]. Grundling et al. [6] estimated that 24 million people are living with T2DM in Africa, and this number is predicted to rise to 55 million by 2045. The impact of T2DM in African countries has reached alarming proportions; hence, Northern Africa (driven by Egypt) and Southern Africa (driven by South Africa) appear to have the highest number of people living with T2DM than the global average [7].

South Africa has approximately 4.58 million people living with T2DM [6]. This figure has nearly tripled from 4.5% in 2010 to 12.7% in 2019. Among other provinces, KwaZulu-Natal has the highest prevalence of T2DM cases [8]. In KwaZulu-Natal (KZN), the figure is 12.5%, which increases to 34.1% when combined with patients with private medical aid [8]. This figure is, however, considered an underestimate as T2DM is frequently undiagnosed in the country [8]. The importance of lifestyle in the development of diabetes mellitus provides an opportunity to promote healthy lifestyle changes as a preventive strategy and a first line of defense against the increasing prevalence of type 2 diabetes mellitus [9]. Despite exercise and physical activity being shown to improve insulin resistance and T2DM, they remain relatively underutilized in comparison to medicinal or pharmaceutical treatments [10]. Physical activity, defined as any bodily movements that raise energy expenditure above resting levels, leads to improvements in insulin sensitivity, body weight, physical fitness, lipid levels, and overall well-being. Additionally, it reduces the risk of cardiovascular morbidity and mortality [11]. In contrast, exercise is defined as a planned, structured, and repetitive form of physical activity aimed at enhancing physical fitness. Nutritional interventions such as consuming correctly classified sources of macronutrients result in good health and weight control. Bradshaw et al. [12] reported that eating habits significantly influence the management of T2DM. When combined with physical activity, they are crucial for weight control [1]. 

Globally, knowledge related to understanding and management of T2DM is limited, despite the critical nature of this disease. Knowledge, defined as skills, information, and facts acquired through education or experience, is a powerful tool in the fight against T2DM. Knowledge forms the basis for informed decisions about eating habits, weight control, blood glucose monitoring, and the use of medications [1,13]. A lack of sufficient diabetes knowledge frequently leads to poor self-care management skills and poor control of blood glucose, which consequently leads to poor compliance with both medication and lifestyle changes [14,15]. A study by Reid et al. [16] in the Free State province of South Africa reported poor knowledge in people with T2DM in areas with high incidence of T2DM. People living in rural areas often have limited knowledge about diabetes due to their educational background, which hinders their access to essential information [17]. A study by Reid et al. [16] highlighted that people living in rural areas often lack the education and resources needed to manage T2DM effectively. Traditionally, people from these regions tend to perceive being overweight or obese as a sign of good health and a better socioeconomic status, which is a misconception [18]. Obesity and overweight are associated with T2DM and heart diseases [7]. Patient education is therefore the cornerstone of care for individuals with T2DM. Sufficient diabetes knowledge enables individuals with T2DM to think positively and structure a mindset that enables a positive attitude toward behavioral change [8,19].

Attitude is defined as an intention and ultimate practice related to T2DM, as expressed by people with T2DM, to internalize diabetes-related information and make good judgments about diabetes self-care and management, which plays a vital role in managing T2DM. A positive attitude in people with T2DM may lead to a significant behavioral change [20]. According to Belsti et al. and Chiwungwe [19,20], the attitude of an individual is largely influenced by numerous factors such as knowledge regarding the disease, family, educational status, residence, and how society defines a person with T2DM. Individuals with T2DM may perceive physical activity as a burden or believe that their condition limits them from engaging in physical activity [20]. Individuals with type 2 diabetes mellitus (T2DM) may fear injury from physical activity or worsening complications associated with the condition, which can contribute to negative attitudes toward exercise [20]. Alternatively, positive attitudes are evident among those who view exercise as an opportunity to improve their overall health and well-being [16]. An individual’s attitude toward T2DM is key in the adoption and maintenance of certain behaviors that are essential in treating and managing T2DM [8]. Therefore, practice plays a significant role in the management of T2DM.

Positive practice towards exercise and its benefits plays an equally important role in the management of T2DM through regular physical activity. Maintaining consistent involvement in physical activity requires overcoming common barriers such as lack of time, limited access to appropriate facilities or resources, social support deficiencies, and negative beliefs about personal abilities or the benefits of exercising [21]. Effective strategies for promoting active lifestyles among people with T2DM involve addressing these barriers by adopting tailored interventions based on individual needs and preferences. Therefore, the study aimed to determine an individual’s T2DM diabetes-related knowledge, attitude, and practice toward exercise and its benefits among individuals with T2DM.

## 2. Materials and Methods

### 2.1. Study Population, Sample Size, and Sampling

A cross-sectional descriptive research design was used to evaluate diabetes-related knowledge, attitude, and practice toward exercise and its benefits among individuals with T2DM. A total of one hundred and ninety-nine participants (199) participated in the study. The study population was individuals with T2DM attending church in KwaZulu-Natal-Durban and community members from Chesterville. The participants were males and females diagnosed with T2DM, aged 18 and above, and literate. Participants who were excluded from the study were those who presented with other diabetic conditions such as T1DM and gestational diabetes mellitus.

### 2.2. Procedure

The validated adapted South African Diabetes-related Knowledge, Attitude, and Practice (KAP) questionnaire was used in the study. The questionnaire was designed based on a pre-validated and reliable questionnaire by Le Roux and Niroomand et al. [18,22] to collect data relevant to knowledge, attitude, and practice in individuals with T2DM about exercise and its benefits. The modified questionnaire was piloted and refined before commencing the study The questionnaire consisted of KAP closed-ended structured questions, evaluated by either true/false/unsure. The questionnaire was categorized into various sections: Section A consisted of information on the socio-demographic distribution of variables of participants, such as age, sex, and ethnic group. Section B covered 10 items on knowledge of T2DM towards exercise and its benefits, assessed with correct and incorrect responses. Knowledge was scored by summing the scores of all questions of the domain. The scores were then converted to percentages, and knowledge was determined by the number of selected correct/incorrect responses. Section C covered 10 attitude questions on T2DM towards exercise and its benefits and Section D covered 10 practice questions on T2DM towards exercise and its benefits. Attitude and practice were assessed using a three-point scale, that is, true, false, and unsure. Some items were reverse-coded because of the negative meanings of the items. A score above 50% was categorized as either a poor or good attitude/practice, depending on the question asked.

### 2.3. Ethical Considerations

Ethical approval was obtained from the University of KwaZulu-Natal (UKZN) Biomedical Research Ethics Committee (BREC/000033892/2021) and permission to conduct the study was obtained from the church. Written informed consent was obtained from participants before enrolling in the study.

### 2.4. Data Analysis

The collected data were computed on an Excel spreadsheet and analyzed using SPSS version 23.3.1. Descriptive statistics such as percentages for categorical data, frequencies, means, and standard deviations (SDs) were calculated and used to summarize the demographic profiles of the study participants. The results are presented in tables and graphs, and the association between variables was determined using appropriate statistical interpretations. The chi-square test was used to determine the association between knowledge and age. When conditions were not met, Fisher’s exact test was used. Hence, a probability value (*p*-value) of less than 0.05 was accepted as statistically significant. 

## 3. Results

### Demographic Characteristics of the Participants

The demographic characteristics of the study participants are presented in Figure 1. A total of 199 individuals with T2DM met the inclusion criteria and were invited to partake in the study. The mean age of the study participants was 58.65 (SD ± 12.04) years and more than half of the study participants aged between 60 and 69 (48.80 ± 16.64) participated in the study, with under 30, 57.50 (SD ± 17.53), and 30–39, 42.50 (SD ± 13.89) being the lowest. Overall, 78.4% were female and 21.6% were male participants. The majority of the study participants were Black African (95.5%), followed by White participants, (3.0%) and the colored population made up only 1.5% of the study participants. 

Table 1 showed that the majority 163 (82%) of individuals with T2DM did not know that physical activity and exercise are different (*p* < 0.001), followed by 150 (75%) individuals, indicating that a person with T2DM cannot do strenuous exercise like weightlifting, cycling, and running (*p* < 0.001). More than 50% of individuals with T2DM (134 (67%)) indicated that herbs make a person with T2DM healthier than if they use Western medication (*p* < 0.001). The majority (139 (70%)) believed that a person with T2DM using any type of medication can be cured of the disease (*p* < 0.001).

The results (see Table 2 below) showed that the vast majority (158 (79.4%)) of individuals with T2DM felt that their regular work is an adequate substitution to exercise (*p* < 0.001), followed by 128 (64.3%) of the study participants indicating that they need someone to keep prompting them to do exercises (*p* < 0.001). The majority (139 (69.8%)) of the participants indicated that they use mild pain or fatigue as an excuse to keep them away from exercising (*p* < 0.001). Among 199 of the study participants, 125 (68.3%) indicated that they do not look forward to doing exercises (*p* < 0.001), with 137(68.8%) indicating that they felt that age is an influencing factor in motivating them to do exercises (*p* < 0.001).

The overall results (Table 3 below) showed a significant positive practice towards exercise and its benefits among individuals with T2DM. The results in Table 3 showed that 157 (78.9%) of the participants participated in activities that make them sweat as they exercise (*p* < 0.001), with 141 (70.9%) of the study participants indicating that they found out how they can adjust their lifestyle to living with T2DM (*p* < 0.001). In total, 124 (62.3%) of the study participants check their blood pressure monthly (*p* < 0.001), and 136 (68.3%) reported that they do ask therapists as to how they can perfectly learn exercises (*p* < 0.001). 

## 4. Discussion

The predominance of females (78.4%) in the present study aligns with the findings of Reid et al [16]. Females are more likely to report obstacles to exercise and feel less control over their decision to engage in physical activity, which can negatively impact their knowledge and attitudes toward exercise and its benefits in managing T2DM. In contrast, males tend to be more physically active. Traditional female gender roles, such as childcare and housework, may contribute to lower participation in physical activity. These roles can also make it challenging for women to prioritize their health.

The majority of the respondents showed poor knowledge among individuals with T2DM towards exercise and its benefits, and this concurred with the results of Mwimo et al. [23] in Tanzanians, which reported minimal knowledge about exercise in managing T2DM. However, it is essential to recognize that participating in physical activities may vary based on the study population and specific contextual factors. The inability of individuals to manage T2DM effectively contributes to the high rate of physical inactivity in the country. Studies have shown that individuals with T2DM often lack knowledge about their condition, leading to inadequate self-management practices, poor glycemic control, and an increased risk of complications. According to Odili et al. [24], a lack of knowledge about T2DM is associated with an increased rate of hospitalization.

A study by Okolie et al. [25] reported poor knowledge among individuals with T2DM. More than half (67%) of the study’s participants reported the use of traditional medicine (herbs) to cure T2DM, and this concurred with the study by Okolie et al. [25], which reported the use of herbs as a cure for T2DM; however, this may interfere with orthodox treatment. Traditional medicine encompasses practices rooted in the theories, beliefs, and experiences of various cultures. The widespread use of African traditional medicine by nearly 72% of the Black African population before the introduction of orthodox medicine has been well documented. Several studies confirm that cinnamon, ginger, fenugreek, bitter gourd, ivy gourd, and crepe ginger are among the most commonly used herbal remedies for T2DM [26,27]. 

In South Africa, popular herbal remedies for T2DM include *Vernonia amygdalina* (bitter leaf), *Hypoxis hemerocallidea* (African potato), *Mimusops zeyheri*, *Catharanthus roseus* (Madagascar periwinkle), and *Sutherlandia frutescens* (cancer bush) [26]. The biological and pharmacological effects of *Aloe ferox*, *Artemisia afra*, and *Leonotis leonurus* have been extensively studied. *Aloe ferox* improves carbohydrate metabolism and reduces obesity-related glucose intolerance, while *Artemisia afra* and *Leonotis leonurus* have demonstrated hypoglycemic and hypolipidemic effects [26]. A study by Mukeshimana and Nkosi [28] focused on the Rwandan population and found that many individuals, particularly among the Black community, still believe that T2DM can only be treated with traditional medicine. The researchers argued that Rwandans often perceive a person exhibiting symptoms related to T2DM as being bewitched, which leads them to seek help from traditional healers and to use traditional remedies as the preferred method for addressing the illness [28]. 

The study indicated a significant relationship between age and knowledge of type 2 diabetes mellitus (T2DM). The results showed that participants aged 60–69 had notably higher knowledge, with an average score of 42.80 (SD ± 15.64), compared to other age groups. This is mainly due to experience and increased information seeking. Older adults are more likely to seek information about T2DM compared to young adults [29]. The study findings concurred with the results by Aljofan [29], which reported that older age groups are associated with high diabetes knowledge. With all the myths, disbelief, and lack of knowledge about T2DM, it is thus evident that individuals still lack knowledge about exercise and its benefits. 

The study results also demonstrated negative attitudes towards exercise and its benefits in individuals with T2DN, and this concurs with the study by Imam and Dharepgol [30], which found that 80% of individuals with T2DM demonstrated negative attitudes towards exercise. A study by Alaofe et al. [31] indicated a relatively negative attitude in individuals with T2DM. Individuals with T2DM reported that stigma contributed to their negative attitudes and impacted their psychological well-being. Moreover, factors such as not adhering to a healthier lifestyle and taking medication are contributing factors to negative attitudes. Similar findings were reported in an Ethiopian study involving patients over 50 years of age with T2DM [31]. More than 79.4% of the study participants indicated that their work can be used as a substitute for exercise and this is consistent with the findings of Sookan et al. [32], which indicated that participants viewed the daily activity as similar to exercise, which is untrue. Furthermore, only 29% practiced regular structured exercises as per international recommendations. These results indicate that there is a significant gap between knowledge and practice regarding the role of exercise in T2DM management.

Despite the higher level of poor knowledge and negative attitude towards exercise and its benefits among individuals with T2DM reported in this study, participants demonstrated good practice towards exercise and its benefits. The study participants showed good practices towards preventative measures against T2DM, and this is consistent with the results of Almousa et al. [21], which indicated that the highest behavior displayed by Saudi Arabians was the commitment to take medication according to the doctor’s instructions. This includes checking blood sugar every month (62.3%), seeking out information on how to live with T2DM (70.9%), and checking for foot injuries regularly (65.3%). The findings of the study are consistent with the results of the study by Reid et al. [16], which reported that patients with T2DM regularly take their medication. 

## 5. Conclusions

The study revealed that participants had a poor understanding and attitude toward exercise and its benefits, even though their actual practices were commendable. Alarmingly, most participants still relied on traditional medicine for managing and preventing T2DM. These misconceptions can significantly impact the management of T2DM within the community. As the number of individuals diagnosed with T2DM continues to rise, it is increasingly important to understand and address the factors that contribute to the effective management and the prevention of complications associated with this condition. As part of a comprehensive diabetes care plan, healthcare providers should actively encourage and educate individuals with T2DM on being physically active. By promoting this holistic approach to diabetes management, healthcare professionals can empower patients to take control of their condition. The integration of multidisciplinary teams, particularly nursing staff, is vital in implementing these educational initiatives successfully. Such interventions contribute significantly to better management of T2DM and a reduction in associated health complications.

## Figures and Tables

**Figure 1 ijerph-21-01529-f001:**
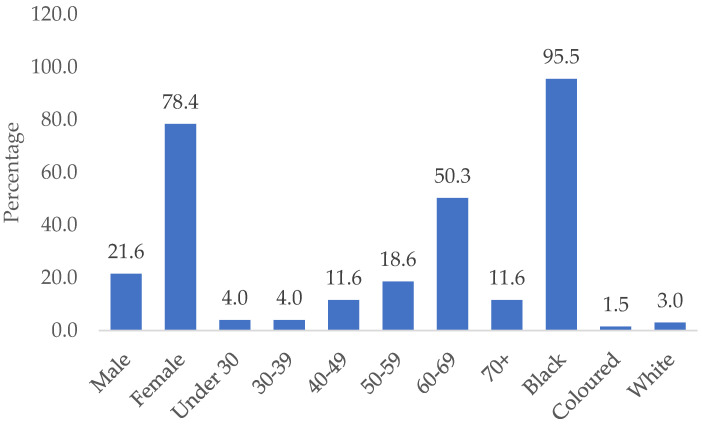
Distributions of socio-demographic characteristics of individuals with T2DM (*n* = 199).

**Table 1 ijerph-21-01529-t001:** Participants’ T2DM-related knowledge (*n* = 199).

Item	Frequencies (%)	
	Correct	Incorrect
Physical activity and exercise are the same thing?	36 (18)	163 (82)
People with T2DM can safely perform exercise?	140 (70)	59 (30)
Exercise can be used in the management/treatment of T2DM.	138 (69)	61 (31)
A person with T2DM cannot do strenuous exercise like weightlifting, cycling, or running.	49 (25)	150 (75)
I am willing to engage in exercise to improve my health.	131(66)	68 (34)
Type 2 diabetes mellitus management should include both exercise and a healthy diet.	134 (67)	65 (33)
A person with T2DM will often have high blood pressure.	61 (31)	138 (69)
A person with T2DM using herbs makes that person healthier than if they use Western medication.	65 (33)	134 (67)
A person with T2DM using any type of medication be cured of the disease?	60 (30)	139 (70)
Type 2 diabetes mellitus medication may cause swelling of the feet.	57 (29)	142 (71)

**Table 2 ijerph-21-01529-t002:** Participants’ T2DM-related attitude (*n* = 199).

Item	Responses as Frequency (%)		
	True	False	Unsure
I feel that my regular work is an adequate substitute for exercise	158 (79.4)	37 (18.6)	4 (2.0)
I need someone to keep prompting me to do my exercises	128 (64.3)	58 (29.1)	12 (6.0)
I use mild pain or fatigue as an excuse to keep away from my exercises	139 (69.8)	55 (27.6)	5 (2.5)
I will continue my exercises until I improve, regardless of how long it takes	49 (24.6)	145 (73.9)	5 (2.5)
I believe I will improve with exercises as I have seen others improve	59 (29.6)	129 (68.4)	11 (5.5)
I look forward to doing my exercises each day	60 (30.2)	125 (63.8)	14 (7.0)
I feel that age is an influencing factor in motivating me to do my exercises	137 (68.8)	57 (28.6)	5 (2.5)
I feel embarrassed doing exercise in front of others	128 (69.3)	63 (31.7)	8 (4.0)
I feel that I have no time of my own and my daily exercises take away my valuable time	130 (65.3)	63 (31.7)	6 (3.0)
I give up on exercises owing to the difficulty in sticking to a schedule	128 (64.3)	65 (37.7)	6 (3.0)

**Table 3 ijerph-21-01529-t003:** Participants’ T2DM-related practice (*n* = 199).

Item	Responses as Frequency (%)		
	True	False	Unsure
I do activities that make me sweat as I exercise	157 (78.9)	38 (19.1)	4 (2.0)
I find out how I can still adjust my lifestyle to living with T2DM	141 (70.9)	46 (23.1)	12 (6.0)
I check my blood pressure monthly	124 (62.3)	58 (29.1)	17(8.5)
I check my feet for injuries regularly	130 (65.3)	57 (28.6)	12 (6.0)
I find it difficult to lose weight if I become overweight	133 (66.8)	57 (28.6)	9(4.5)
I feel that my therapist is making tall claims when he explains to me the benefits of the exercise program	136 (68.3)	53 (26.6)	10 (5.0)
I thought of asking my doctor if there are any medicines available, which will make me better, without doing exercise	131 (65.8)	58 (29.1)	10 (5.0)
I keep asking my therapist how perfectly I have learned the exercises or how better I could do it	133 (66.8)	55 (27.6)	11 (5.5)
I am prompt in doing my exercises regularly as it keeps me alert and energetic throughout the day	139 (69.8)	55 (27.6)	5 (2.5)

## Data Availability

The anonymized data will be available on request for those interested due to ethical considerations.

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
