# Peer review of "Diabetes-Related Knowledge, Attitude, and Practice Towards Exercise and Its Benefits Among Individuals with Type 2 Diabetes Mellitus"

_ijerph, 2024, doi:10.3390/ijerph21111529_

Round 1

Reviewer 1 Report

Comments and Suggestions for Authors

The MS entitled "Diabetes-Related Knowledge, Attitude, and Practice Towards Exercise and Its Benefits Among Individuals with Type-2 Diabetes Mellitus" has importance in the study. However this reviewer has certain queries.

Comments

1. There is confusion between "physical activity" and "exercise." Defining these terms clearly in the introduction would eliminate ambiguity.

2. Clarify the distinction between physical activity and exercise. Defining these terms in the introduction or methodology will help readers understand the relevance of this confusion in the study results.

3. A deeper exploration of how cultural beliefs, like traditional medicine, affect diabetes management and exercise adoption would strengthen the discussion and aid in developing effective interventions.

4. While the manuscript discusses the causes of diabetes, it should mention non-enzymatic glycation as a key contributor. Additionally, the following references should be cited:

Ahmad et al., "Aldose reductase inhibitory and antiglycation properties of phytoconstituents of Cichorium intybus: Potential therapeutic role in diabetic retinopathy." International Journal of Biological Macromolecules (2024).

5. Given the poor knowledge and attitudes toward exercise, suggesting educational or behavioral interventions for healthcare providers would be beneficial.

6. Since 78.4% of participants were female, discussing how gender influences exercise-related knowledge and practices could provide valuable insights.

7. The cross-sectional design limits the ability to assess changes over time. Longitudinal data would give a more comprehensive view of these factors.

Comments on the Quality of English Language

English language can be improved in the revised paper.

Reviewer 2 Report

Comments and Suggestions for Authors

It is difficult to dispute the results obtained in this study due to the particular nature of the sample, which is deeply influenced by a deeply rooted and distinctive ethnic, cultural and social component, in this case related to South Africa. The specific characteristics of this population, in terms of educational level, cultural background and socio-economic context, make it difficult to generalise the results.

When comparing these results with those from other cultural and educational contexts, such as Rwanda, Saudi Arabia or Tanzania (East Africa), a significant bias in the results becomes evident. This bias is mainly due to the fact that the central variable in the study is the level of situational knowledge, which is closely linked to the educational and economic opportunities available in each society. Societies with less access to economic resources and less educational development tend to show a reduced level of knowledge, which has a direct impact on the results of the study, thus limiting its applicability to other cultural and social contexts.

Round 2

Reviewer 2 Report

Comments and Suggestions for Authors

Corrections are made to the previously reviewed suggestions in order to give the study rigour and criteria.